# Parameters of Oxidative Stress in Reproductive and Postmenopausal Mexican Women

**DOI:** 10.3390/ijerph17051492

**Published:** 2020-02-26

**Authors:** Araceli Montoya-Estrada, Karla Guadalupe Velázquez-Yescas, Daniela Belen Veruete-Bedolla, José David Ruiz-Herrera, Alma Villarreal-Barranca, José Romo-Yañez, Guillermo Federico Ortiz-Luna, Arturo Arellano-Eguiluz, Mario Solis-Paredes, Arturo Flores-Pliego, Aurora Espejel-Nuñez, Guadalupe Estrada-Gutierrez, Enrique Reyes-Muñoz

**Affiliations:** 1Coordination of Gynecological and Perinatal Endocrinology, National Institute of Perinatology, Ministry of Health, México City 11000, Mexico; ara_mones@hotmail.com (A.M.-E.); kargpe_veye@hotmail.com (K.G.V.-Y.); daniela.veruete@gmail.com (D.B.V.-B.); dr.david.inper@gmail.com (J.D.R.-H.); alma.vvb@gmail.com (A.V.-B.); jryz@yahoo.com (J.R.-Y.); 2Peri and Postmenopause Clinic. National Institute of Perinatology, Ministry of Health, México City 11000, Mexico; g_ortiz_luna@hotmail.com (G.F.O.-L.); artedro@hotmail.com (A.A.-E.); 3Department of Human Genetics and Genomics, National Institute of Perinatology, Ministry of Health, México City 11000, Mexico; msolis_83@yahoo.com.mx; 4Department of Immunobiochemistry, National Institute of Perinatology, Ministry of Health, México City 11000, Mexico; arturofpliego@gmail.com (A.F.-P.); auro.espejel@gmail.com (A.E.-N.); 5Direction of Research, National Institute of Perinatology, Ministry of Health, México City 11000, Mexico; gpestrad@gmail.com

**Keywords:** reactive oxygen species, postmenopause, protein damage, total antioxidant capacity, malondialdehyde

## Abstract

In the reproductive phase, women experience cyclic changes in the ovaries and uterus, and hormones regulate these changes. Menopause is the permanent loss of menstruation after 12 months of amenorrhea. Menopause is also linked to a decrease in estrogen production, causing an imbalance in oxidative stress. We aimed to compare the three stages of lipid peroxidation, protein oxidative damage, and total antioxidant capacity (TAC) between reproductive-aged women (RAW) and postmenopausal women (PMW) in Mexico. We carried out a cross-sectional study with 84 women from Mexico City, including 40 RAW and 44 PMW. To determine the oxidative stress of the participants, several markers of lipid damage were measured: dienes conjugates (DC), lipohydroperoxides (LHP), and malondialdehyde (MDA); exposure to protein carbonyl is indicative of oxidative modified proteins, and TAC is indicative of the antioxidant defense system. Biomarkers of oxidative stress were significantly lower in RAW vs. PMW. DC were 1.31 ± 0.65 vs. 1.7 ± 0.51 pmol DC/mg dry weight (*p* = 0.0032); LHP were 4.95 ± 2.20 vs. 11.30 ± 4.24 pmol LHP/mg dry weight (*p* < 0.0001); malondialdehyde was 20.37 ± 8.20 vs. 26.10 ± 8.71 pmol MDA/mg dry weight (*p* = 0.0030); exposure of protein carbonyl was 3954 ± 884 vs. 4552 ± 1445 pmol PC/mg protein (*p* = 0.042); and TAC was 7244 ± 1512 vs. 8099 ± 1931 pmol Trolox equivalent/mg protein (*p* = 0.027). PMW display significantly higher oxidative stress markers compared to RAW; likewise, PMW show a higher TAC.

## 1. Introduction

The human menstrual cycle involves a highly regulated chain of neuroendocrine phenomena that are initiated and maintained in the arcuate nucleus in the hypothalamus, the master gland for reproduction [1]. The climacteric period is a transition from the reproductive phase to its extinction and cessation of the menstrual function. The climacteric period is characterized by an initial instability of endocrine ovarian function and, subsequently, a drop in ovarian hormone production and the definitive loss of endometrial cycling [1]. Menopause is associated with a decrease in circulating estrogen and progesterone levels, an increase in follicle stimulating hormone levels [2], and the production of reactive oxygen/nitrogen species (ROS/RNS) [3].

Oxidative stress (OS) is a consequence of the use of oxygen in aerobic respiration by living organisms and is characterized as a persistent condition of an imbalance between the generation of ROS and the ability of the endogenous antioxidant system (AOS) to detoxify ROS [4]. The imbalance between pro-oxidants and antioxidants, which may result from diminished levels of antioxidants (AOX) or increased production of ROS [5], leads to tissue damage, lipids, membranes, proteins, and nucleic acids [6].

The main factors associated with an increase in oxidative stress in menopausal women include obesity, aging, decreased estrogen production, and diet [4,7,8,9]. Chronic inflammation is a hallmark of aging and obesity and has been implicated in the release of ROS from different sources, including activated macrophages and neutrophils involved in inflammation that generates oxidants, such as peroxynitrite and nitrosoperoxycarbonate, hypohalous acids, and nitrosating agent [7]. ROS can participate in lipid peroxidation and generate products such as etheno-, propano-, and malondialdehyde that interact with DNA, form DNA adducts, and damage the DNA structure [4,7]. The protective effect of estrogen on oxidative stress is mediated by translocation for specific enzymes from the cytosol that prevent mitochondrial DNA from oxidative attack by free radicals [8]. The mitochondria play an important role in the regulation of cell survival and apoptosis, and the respiratory chain is the primary structural and functional component that is influenced by estrogen activity [8]. The decrease in estrogen levels during menopause leads to changes in the lipid profile [8] and the increase of lipoperoxidation [10]. Estrogen produces direct antioxidant effects by increasing the activities of antioxidant enzymes, such as glutathione peroxidase, and causing an increase in antioxidant vitamin levels. Hence, antioxidants also contribute indirectly to antioxidant capacity [11]. Depletions of dietary antioxidants (long chain fatty acids vitamin C (ascorbic acid), vitamin E (α-tocopherol), vitamin A, β-carotene, metallothionein, polyamines, melatonin, nicotinamide adenine dinucleotide phosphate, adenosine, co-enzyme Q-10, urate, ubiquinol, polyphenols, flavonoids, phytoestrogens, cysteine, homocysteine, taurine, methionine, S-adenosyl-l-methionine, resveratrol, and nitroxides) [5] and other essential dietary constituents (e.g., copper, iron, zinc, and magnesium) can also cause OS [9], whereas overnutrition can lead to increased OS [5,12].

Some authors have shown interest in the biochemical changes at the OS level that occur from the transition stages of reproductive-aged women (RAW) to postmenopausal women (PMW) [6,11,12,13,14,15]. Previous studies reported an increase in levels of OS markers among PMW compared to premenopausal women or RAW.

Total antioxidant capacity (TAC) includes the synergic and redox interactions between the different molecules present in foods and biological fluids [16]. However, the results regarding TAC are not clear; in some studies, TAC was increased [6,14], but in other studies, TAC was decreased in PMW [11,13,15,17].

The lipid peroxidation process is generally described as a process under which oxidants, such as free radicals or nonradical species, attack lipids containing carbon–carbon double bond(s), especially polyunsaturated fatty acids (PUFAs), which involve hydrogen abstraction from a carbon, with oxygen insertion resulting in lipid peroxyl radicals and hydroperoxides [18]. The overall process consists of three steps: initiation, propagation, and termination. In initiation, prooxidants abstract the allylic hydrogen, forming the carbon-centered lipid radical; the carbon radical tends to be stabilized by a molecular rearrangement to form a conjugated diene. In the propagation phase, a lipid radical rapidly reacts with oxygen to form a lipid peroxy radical that abstracts a hydrogen from another lipid molecule, generating a new lipid radical and lipid hydroperoxide. In the termination reaction, antioxidants donate a hydrogen atom to the lipid peroxyl radical species, resulting in the formation of nonradical products [18].

Unlike previous reports, which only reported some biomarkers of oxidative damage without fully describing the process of lipoperoxidation in addition to the antioxidant status. In our study, we aimed to compare the three stages of lipid peroxidation, protein oxidative damage, and TAC between Mexican RAW and PMW.

## 2. Materials and Methods

A cross-sectional comparative study was conducted at the Peri and Postmenopause Clinic of the Instituto Nacional de Perinatología “Isidro Espinosa de los Reyes” in Mexico City, México. The study was performed according to the principles of the Declaration of Helsinki and was approved by the clinic’s Institutional Review Board and registered with number 3210-10209-01-574-17.

### 2.1. Subjects

We included 84 women aged 20 to 60 years who provided written informed consent. Two groups were integrated. The inclusion criteria for each group were Group 1, RAW: age 20–30 years and regular menstrual cycles; Group 2, PMW: 50–60 years old and amenorrhea for at least 12 months. Exclusion criteria were women with replacement hormone therapy, chronic or metabolic diseases, surgical menopause, any pharmacologic treatment in the last three months before admission for the study, tobacco or alcohol consumption, and multivitamin supplements.

### 2.2. Blood Samples

Whole blood 6 mL samples were obtained from an antecubital venipuncture in anticoagulated tubes with 2 mM ethylenediaminetetraacetic acid (BD Vacutainer, Franklin Lakes, NJ, USA) of each participant. The samples were centrifuged at 3000 rpm for 15 min to obtain the plasma and were stored in aliquots at −70 °C until their analysis.

### 2.3. Biomarkers of Oxidative Stress

To determine the OS status of the participants, several markers of lipid damage were measured. The process of lipoperoxidation was quantified in three stages. In the first stage of the lipoperoxidation, known as initiation, dienes conjugates (DC) were quantified. For the second stage of lipoperoxidation, known as propagation, lipohydroperoxides (LHP) were quantified. In the last stage of lipoperoxidation, where aliphatic aldehydes are formed, the most representative product, malondialdehyde (MDA), was quantified.

For protein damage, the exposure to protein carbonyl (PC) and the TAC, as indicative of an antioxidant defense system, were measured.

DC were obtained after extraction with chloroform/methanol (2:1). A spectrophotometric assay was performed at 234 nm [19]. DC was quantified using a molar extinction coefficient ε = 2.7 × 10^4^ M^1^ cm^−1^, and the results are reported as pmol DC/mg dry weight.

LHP were evaluated using the assay conditions described by El-Saadani [20], and MDA was measured using 1-methyl-2-phenylindole. A spectrophotometric assay was executed at 586 nm [21].

The carbonylation of proteins leads to irreversible oxidative damage, which often leads to a loss of protein function, severe oxidative damage, and disease-derived protein dysfunction [22]. Exposures of PC were determined using the dinitrophenylhydrazine method [23]. Briefly, 0.05 mL of plasma was mixed with 0.5 mL 10 mM dinitrophenylhydrazine (DNPH) in 2.5 M HCl (or 2.5 M HCl alone for the blank). Samples were left for 1 h at ambient temperature, then 0.5 mL 20% trichloroacetic was added and centrifuged at 3000 rpm for 5 min at 4 °C. The resulting pellets were rinsed twice by centrifugation with 1 mL 5% trichloroacetic acid. The pellet was washed by centrifugation with 2 mL ethanol/ethyl acetate (1:1) and solubilized in 0.5 mL of 6 M guanidine in 20 mM potassium phosphate, pH 2.3. The carbonyl concentration was determined using the extinction molar coefficient ε = 22 M^−1^ cm^−1^ and expressed as pmol PC/mg protein.

Quantification of antioxidant capacity was evaluated according to a method based on cupric-reducing antioxidant capacity (CUPRAC), using copper (II) and neocuproine reagents. The absorbance was determined at 450 nm. The results were expressed as pmol Trolox equivalent/mg protein. The Trolox is a water-soluble analog of vitamin E [24].

All samples were analyzed in duplicate. Body mass index (BMI) was computed as weight divided by height squared (kg/m^2^).

### 2.4. Sample Size

To find a mean difference between groups using levels of malondialdehyde (MDA), with an α error of 0.01 and β of 0.20, 32 participants were required per group.

### 2.5. Statistics

The data are expressed as mean ± standard deviation. Data were analyzed using Student’s *t*-test with Prism 6.0 (GraphPad, San Diego, CA, USA) and a linear regression model (SPSS V.20, Chicago, IL, USA) was used to adjust age, weight, and BMI as confounding variables. Differences were considered significant when *p* < 0.05.

## 3. Results

A total of 98 women were invited to participate; 14 declined by personal reasons, 84 women accepted (Group 1 RAW, *n* = 40; Group 2 PMW, *n* = 44). Table 1 shows the clinical characteristics of the participants. Age and BMI were significantly higher in PMW. All participants in the RAW group were normal weight and in the group of PMW were: 10 normal weight (22.7%), 19 overweight (43.2%), and 15 obese (34.1%).

Figure 1 shows an increase in DC in PMW (1.70 ± 0.51 pmol DC/mg dry weight) compared to RAW (1.31 ± 0.65 pmol DC/mg dry weight), showing a statistically significant difference (*p* = 0.0032), which represents an increase of 30% of DC in PMW.

LHP concentrations were statistically significant and higher in PMW (11.30 ± 4.24 pmol LHP/mg dry weight) compared to RAW (4.95 ± 2.20 pmol LPH/mg dry weight) (*p* < 0.0001). This difference represents 2.7 times more LHP in PMW, as shown in Figure 2.

Figure 3 shows a statistically significant increase in MDA in PMW (26.10 ± 8.71 pmol MDA/mg dry weight) compared to RAW (20.37 ± 8.20 pmol MDA/mg dry weight) (*p* = 0.0030). This difference represents a 30% increase in MDA in PMW.

Concerning the evaluation of protein damage, Figure 4 shows a statistically significant 16% increase in PC in the PMW group (4552 ± 1445 pmol PC/mg protein) compared to the RAW group (3954 ± 884 pmol PC/mg of proteins) (*p* = 0.042).

TAC was statistically significantly increased in PMW (8099 ± 1931 pmol of Trolox equivalent/mg of protein) compared with RAW (7244 ± 1512 pmol of Trolox equivalent/mg of proteins) (*p* = 0.027). This difference represents a 14% increase in PMW (Figure 5).

Table 2 shows the results of the linear regression model adjusted for age, weight, and BMI. Age was significantly associated with the concentration of DC and LHP, whereas weight and BMI were significantly associated with the concentration of PC, but we found no significant association with age. Neither age nor weight were significantly associated with the TAC and MDA concentrations.

## 4. Discussion

In the present study, we found that Mexican PMW had higher levels of OS markers compared to RAW. Postmenopausal age was a determinant factor in both DC and LHP, which are markers of initiation and propagation stages of lipid peroxidation. Weight and BMI were significantly associated with the concentration of PC.

The products of lipoperoxidation, such as 4-hydroxy-2-nonenal, 2-hexenal, and MDA, can react with the amino group of lysine, the imidazole entity of histidine, or the sulfhydryl group of cysteine, forming reactive carbonyl groups that cause protein oxidation via secondary reactions [25]. Therefore, if the concentration of MDA increases, it is possible that protein damage occurs due to exposure to carbonyl groups.

Few studies have reported sensitive and new methodologies that capture the process of complete lipid peroxidation as well as the counterpart of oxidative damage, which, in this case, is antioxidant defense in Mexican women in the postmenopausal stage.

Similar to our findings, Taleb-Belkadi et al. reported that, among 117 Algerian women, the concentrations of thiobarbituric acid reactive substances and carbonylated proteins were higher in peri- (*n* = 40) and postmenopausal (*n* = 47) women compared to RAW (*n* = 30). Contrary to our results in relation to TAC, others reported that superoxide dismutase and catalase activity are lower in PMW compared to RAW, which reflects a decrease in TAC among PMW [11,13,15]. Similarly, Koleniskova et al. [6] observed a decrease in TAC among Russian climacteric women, probably due to the decrease in postmenopausal estrogen levels; that outcome is contrary to our findings. These results could be explained by ethnicity as well as by the enzymatic determinations used to report the TAC, which was different from our study. Some authors had reported higher OS markers in African Americans than [26,27]. Ethnic differences in the distribution of genetic polymorphisms could also contribute to differences in oxidative stress [27]. Additionally, access to health care, socioeconomic status, stress, physical activity, diet, and obesity could explain some ethnic differences in OS status [26,27].

In a study comprising 51 Turkish women (32 premenopausal women and 19 menopausal women), Demirbarg et al. [11] found that TAC was significantly lower in women with lower estradiol [11]. Our lipid damage results are similar to those reported by Sanchez-Rodríguez [28], although different techniques were used for the measurements. In 187 perimenopausal women from Mexico City (94 premenopausal and 93 postmenopausal), significantly higher levels were found in the PMW group. However, the authors reported no significant differences in TAC status between groups.

The measurement of TAC reflects the anti-oxidative status of an organism [29]. Our study showed that the TAC of PMW is 14% higher than RAW; similar findings were reported by Victorino et al. [14] in Brazilian women (30 RAW and 28 PMW), where the TAC increased 1.8 times in PMW; however, OS markers were lower in PMW than RAW, which is contrary to our findings.

The higher oxidative status of postmenopausal women, despite an increase in biomarkers of oxidative damage in PMW, has an antioxidant defense mechanism capable of neutralizing excess free radicals, thus increasing women’s TAC. The organism has several antioxidant defenses to protect against hostile oxidative environments, including classical antioxidant enzymes, such as catalase, glutathione peroxidase, and superoxide dismutase, as well as non-enzymatic ROS scavengers, such as β-carotene, vitamin C, vitamin E, and uric acid [30]. Changes in the redox balance considerably impact the transcriptional activities and cellular signaling pathways because most of the activation and reactions are dependent on the reduction/oxidation processes [31]. We hypothesized that the higher TAC concentration in PMW is due to a natural response to decrease OS markers; however, it requires testing in future studies that measure concentrations of different antioxidant enzymes.

Some authors suggested that menopause could be considered a risk factor for OS [6], as well as a potentially increased risk of cardiovascular diseases [26,27] and osteoporosis [17], due to a depletion of estrogen [32]. The depletion of estrogen in postmenopausal women was suggested to cause OS and increase damage at the mitochondrial and nuclear DNA levels [33,34,35]. Extensive observations suggest that DNA damage accumulates during aging as a consequence of an increase in OS and a decline in DNA repair capacity [35,36]; however, aging of the DNA also results in less expression of genes related to antioxidant defense, which contributes, in part, to the increase in OS observed in PMW as a consequence of aging, and not only caused by ovarian senescence and estrogen depletion [37,38].

Several studies proposed that a supplementary diet with antioxidants could decrease levels of OS markers, and thus, the incidence of comorbidities [33,39]. In the future, randomized clinical trials should evaluate the efficacy of interventions like long chain fatty acids, low-carbohydrate soy diet, resveratrol, fructose restriction, Mediterranean diet, supplementation with resveratrol, niacin, and vitamins E, C, and A, and hormonal replacement therapy to improve OS and antioxidant defenses in PMW [12,36,40].

Although, in the bivariate analysis, TAC and MDA concentrations were significantly higher in PMW than in RAW, and in the linear regression model, neither age nor weight nor BMI contributed significantly to the TAC and MDA concentrations. These differences found in the bivariate analysis could be explained by other factors that were not measured in the present study.

The main strength of this study is that the measurement of the three stages of lipid peroxidation enabled the evaluation of the evolution of the complete process, as well as protein damage.

This study had some limitations: the study design, the lack of control of some variables like diet, estrogens concentration, and markers of metabolic status. Another limitation was that our sample only included two groups of patients; it could also include more age ranges. So, our findings should be interpreted with caution.

## 5. Conclusions

In conclusion, the findings indicate a higher oxidative marker in PMW compared to RAW; markers of initiation and propagation stages of lipid peroxidation are ascribable to age. However, the differences in the termination stage of lipid oxidation and protein damage markers could be attributable not only to age but also to body mass index. Likewise, PMW showed a more active antioxidant defense system.

## Figures and Tables

**Figure 1 ijerph-17-01492-f001:**
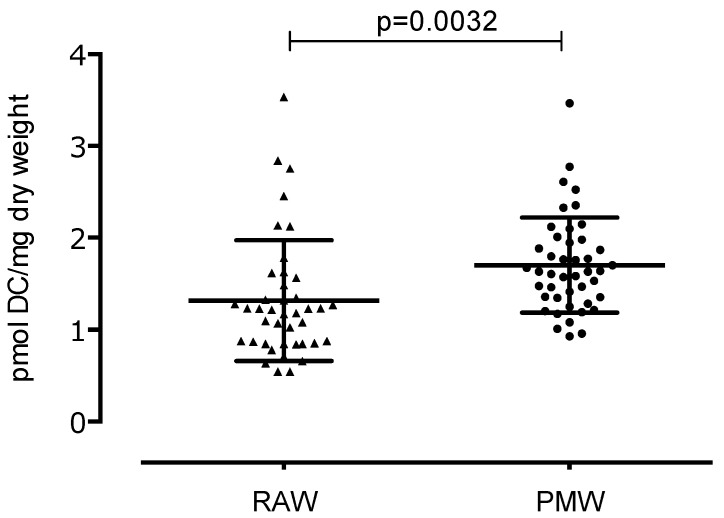
Concentration of the first stage of lipoperoxidation: dienes conjugates in reproductive-aged women (RAW) and postmenopausal women (PMW).

**Figure 2 ijerph-17-01492-f002:**
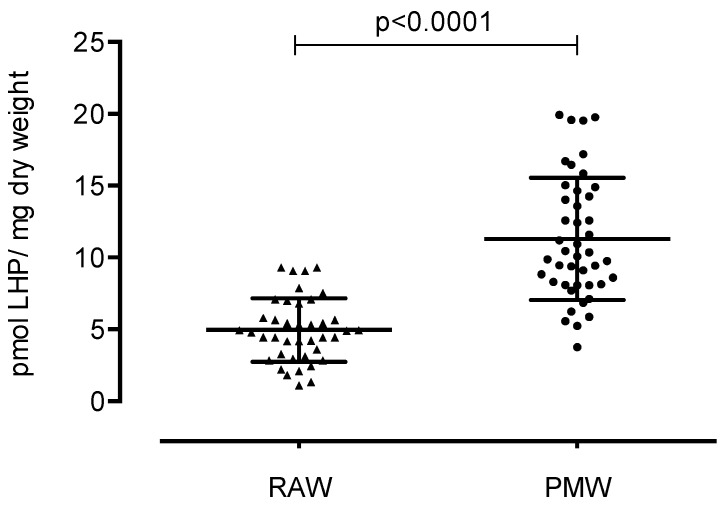
Values of the second stage of lipoperoxidation in reproductive-aged women (RAW) and postmenopausal women (PMW).

**Figure 3 ijerph-17-01492-f003:**
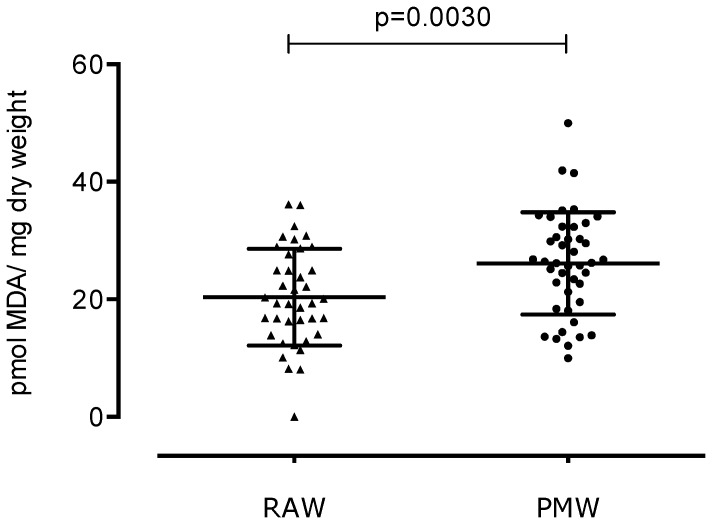
Levels of malondialdehyde (MDA) in reproductive-aged women (RAW) and postmenopausal women (PMW).

**Figure 4 ijerph-17-01492-f004:**
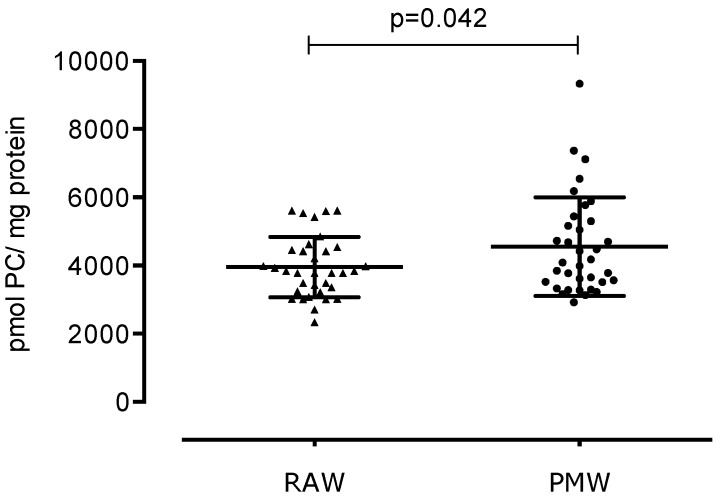
Marker of damage to proteins: Protein carbonylation in reproductive-aged women (RAW) and postmenopausal women (PMW).

**Figure 5 ijerph-17-01492-f005:**
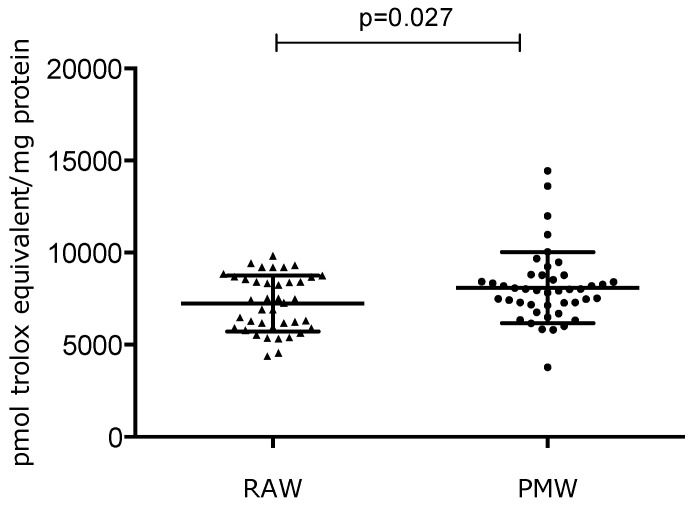
The antioxidant defense evaluated in reproductive-aged women (RAW) and postmenopausal women (PMW).

**Table 1 ijerph-17-01492-t001:** Demographic and clinical characteristics of study participants.

	RAW (*n* = 40)	PMW (*n* = 44)	*p* Values
Age (years)	25.4 ± 0.27	55.06 ± 3.69	*p* < 0.0001
Weight (kg)	52.2 ± 2.14	68.9 ± 11.3	*p* < 0.001
BMI (kg/m^2^)	22.3 ± 1.80	28.7 ± 4.60	*p* < 0.001

RAW: Reproductive-aged women; PMW: postmenopausal women; BMI: body mass index (weight kg/height m^2^); data are presented as means ± SD.

**Table 2 ijerph-17-01492-t002:** Linear regression model adjusted for age, weight, and body mass index (BMI) in oxidative stress.

	Beta	t	*p* Values
DC			
Age (years)	0.298	2.036	0.045
Weight (kg)	−0.105	−0.328	0.744
Body mass index (kg/m^2^)	0.114	0.357	0.722
LHP			
Age (years)	0.57	5.045	<0.001
Weight (kg)	−0.020	−0.082	0.935
Body mass index (kg/m^2^)	0.169	0.683	0.496
MDA			
Age (years)	0.165	1.103	0.274
Weight (kg)	−0.013	−0.038	0.97
Body mass index (kg/m^2^)	0.117	0.358	0.721
PC			
Age (years)	0.113	0.711	0.48
Weight (kg)	0.816	2.358	0.021
Body mass index (kg/m^2^)	−0.674	−1.999	0.05
TAC			
Age (years)	0.167	1.123	0.265
Weight (kg)	0.407	1.123	0.265
Body mass index (kg/m^2^)	−0.348	−1.068	0.289

Note: DC: dienes conjugates; LHP: lipohydroperoxides; MDA: malondialdehyde; PC: exposure to protein carbonyl; TAC: total antioxidant capacity.

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
