# Peer review of "Parameters of Oxidative Stress in Reproductive and Postmenopausal Mexican Women"

_ijerph, 2020, doi:10.3390/ijerph17051492_

Round 1
Reviewer 1 Report
Oxidative stress in reproductive and postmenopausal Mexican women
Montoya-Estrada, A. et al.
Overall summary: This manuscript describes a clinical trial involving women at pre- and post-menopausal status. The authors investigate the degree of oxidative stress in these women caused by elevated reactive oxygen species. It is assumed that the authors hypothesize that post-menopausal women will show more damage due to decreased circulating estrogen concentrations. Previous studies report that estrogen regulates anti-oxidant defenses. The authors also distinguish their study by measuring initiation, propagation, and termination of lipoperoxidation. Regarding measurements of oxidative damage, this study is comprehensive but fails to include subjects’ estrogen concentrations to correlate (or not) with their other parameters. The authors also do not include cellular markers of oxidative stress available in blood, such as DNA, or protein damage in PBMCs. This is not a criteria for rejection, but the authors are encouraged to provide at least estrogen concentrations and statistical correlations with their existing data; this would prove their stated mechanism for elevated ROS. If this is impossible, please tell us why in the limitations section of your discussion. Their provided status of the subjects’ oxidative damage is comprehensive, and it seems that profiling the process of damage from blood parameters is the main hook to this study. If so, please flesh this out in the introduction and highlight this in the conclusions as the defining aspect of the manuscript.
Introduction: Please expand this section to flesh out your proposed mechanism, namely that estrogen decline leads to elevated oxidative stress. How does this happen? Can it be corrected with estrogen supplementation? Secondly, please expand your description of your measurements. Capturing the 3 phases of lipoperoxidation is novel, right? A linear explanation of this process, along with details of your other parameters, is necessary. What does each marker mean for cellular physiology?
Methods: Please report the age range of Group 1. In the description of the CUPRAC assay, please define what Trolox is.
In the Statistics section, line 24, did you mean “confounding” instead of “confusing”? What variables were these?Results: Your description of the 3 phases of lipoperoxidation should be moved to the introduction and broadened.
Table 2 is the perfect place for estrogen concentrations to go. In the LHP section, please replace the P value of 0.000 with P<0.001.
Discussion/Conclusions: Please explain why you think estrogen leads to a climate of oxidative stress, but only two of your parameters were associated with age. Shouldn’t all of them be associated with age?
Your conclusions in the first paragraph are not supported by the data, since not all of your measurements correlate with age. The idea of ethnicity playing a role in your results is fascinating and should be discussed more here. Also, please expand your discussion of TAC. What role might estrogen play? Then sentence on lines 30-31 makes no sense. Why would endogenous defenses create barriers to oxidant reduction? In the conclusion section, please highlight your unique measurements. That is why your study is novel, right?
Author Response
Consulte el archivo adjunto.

Reviewer 2 Report
Article:
The manuscript “Oxidative Stress in Reproductive and Postmenopausal Mexican Women” deals with an interesting study on the oxidative stress and total antioxidant capacity between reproductive age and postmenopausal women. Menopause is associated with the production of reactive oxygen/nitrogen species (ROS/RNS), which suggests a link to oxidative stress in individuals with a deficient antioxidant capacity.
However, a major revision is necessary for getting the acceptance of this paper following the comments below.
Minor comment:
The English language should be still improved to ensure that an international audience can clearly understand your text.
Major comments:
Introduction
I suggest that you can improve about the general knowledge of total antioxidant capacity between reproductive age and postmenopausal women to provide more justification for your study. It would be interesting to add studies more recent (except for Algerian, Turkish and Brazilian women in the discussion section). The references in this section are not sufficient, and there are no recent references.
Materials and Methods
Materials and Methods section is rather unclear, they must be improved.
Discussion
At the line 14:
The findings from this study could support intervention with antioxidants, as part of the diet or as supplements in postmenopausal women.
Have the authors tested specific antioxidants, for example in vitro conditions? Or considered volunteers/patients with a specific diet?
The authors reported at the line 21-22:
Our study showed that the TAC of PMW is 14% higher than RAW. And then, No significant differences in TAC status between groups.
The authors also reported at the line 26 that the higher oxidative status of postmenopausal women, despite an increase in biomarkers of oxidative damage in PMW, has an antioxidant defense mechanism capable of neutralizing excess free radicals, thus increasing women’s TAC.I can’t understand, it must be explained better.
At the line 28:
This finding suggests that, in postmenopause, the antioxidant systems could increase TAC, antioxidant enzymes, and antioxidant co-substrates, while endogenous and exogenous antioxidants generate barriers to reduce oxidants and diminish their oxidative power to confront diverse species and oxidative forms capable of attacking the organism. But there are no experiments about the levels of exogenous antioxidant enzymes or antioxidant co-substrates in both groups. There are no specific references in this section.
At the line 33:
Some authors have suggested that menopause could be considered a risk factor for OS, as well as a potentially increased risk of cardiovascular diseases and osteoporosis, due to a depletion of estrogen. Several studies have shown that a supplementary contribution of antioxidants decreases levels of OS markers, and thus the incidence of comorbidities. There are no references about the final phrase.
And I can't understand your conclusions. I am agree with the authors, this study has some limitations; the study design, the lack of control of some variables like diet, estrogens concentration and markers of metabolic status. So, our findings should be interpreted with caution. Another limitation is that your study only included two groups of patients; it could also include in more age ranges.
Reviewer 3 Report
The manuscript submitted by Montoya-Estrada and collaborators is an interesting study. The study has an adequate structure and writing, the introduction is good, the methodology used is sufficient, the results are interesting and support the discussion. However I have the following comments.
Major Comments:
1. Improve the wording of the inclusion and exclusion criteria.
2. In a small number of participants, but the authors could in table 1, add to BMI, the% of women who are normal, overweight, obese or weak.
3. The introduction is interesting, but I suggest including a brief paragraph regarding the role of aging, obesity and diet (unhealthy diet), in the development of oxidative stress.
4. It would be very interesting to include results of antioxidant enzyme activity (for example in erythrocytes), and levels of GSH (total, oxidized and reduced) in the study. I suggest including them, only if possible.
5. The discussion is interesting and novel, however I suggest including obesity and diet (specific components such as saturated fat, sucrose, fructose of industrial origin, etc.) as factors that favor the development of oxidative stress.
Suggested References:
Valenzuela & Videla. The importance of the long-chain polyunsaturated fatty acid n-6 ​​/ n-3 ratio in development of non-alcoholic fatty liver associated with obesity. Food Funct 2011; 2: 644-8.
Hernandez-Rodas et al., Relevant Aspects of Nutritional and Dietary Interventions in Non-Alcoholic Fatty Liver Disease. Int J Mol Sci. 2015; 16: 25168-98.
6. The increase in oxidative stress would only be a consequence of a decrease in systemic antioxidant defenses ?, or is it likely that aging of the DNA produces less expression of genes related to the antioxidant defense ?, briefly discuss this point.
Minor comments:
1. Review the writing of the manuscript, some sentences are very extensive.
2. Improve the wording of the objective of the study.
3. In the title I suggest to include "Parameters of oxidative stress ......."
Round 2
Reviewer 1 Report
The manuscript is much better now with the edits you made. Your novelty is highlighted and conclusions expanded. Nice work.
Reviewer 2 Report
Minor revision is necessary for the manuscript "Parameters of Oxidative Stress in Reproductive and Postmenopausal Mexican Women".
English language and style are fine, only general spell check required.
References are sufficient, but they must be corrected in the style of the journal.